Sixteenth-century tomatoes in Europe: who saw them, what they looked like, and where they came from

van Andel Tinde tinde.vanandel@naturalis.nl 1 2 3
Vos Rutger A. 2
Michels Ewout 2 3
Stefanaki Anastasia 1 2
1 Biosystematics Group, Wageningen University and Research , Wageningen , The Netherlands
2 Naturalis Biodiversity Center , Leiden , the Netherlands
3 Institute for Biology, Leiden University , Leiden , the Netherlands
Calderini Ornella
Electronic publication date: 2022 Jan 17
Publication date: 2022
Volume: 10
Electronic Location ID: e12790
Received 2021 Jul 9; Accepted 2021 Dec 22
Copyright: ©2022 van Andel et al.
Copyright year: 2022
Copyright holder: van Andel et al.
License: This is an open access article distributed under the terms of the Creative Commons Attribution License, which permits unrestricted use, distribution, reproduction and adaptation in any medium and for any purpose provided that it is properly attributed. For attribution, the original author(s), title, publication source (PeerJ) and either DOI or URL of the article must be cited.
License URL: https://creativecommons.org/licenses/by/4.0/

Keywords: Tomato, Renaissance, Historical herbaria, Botanical illustrations, Colonial history, Solanum lycopersicum, 16th century, Landraces, Crop diversity, New World crops

Funding: Naturalis Biodiversity Center This research was funded by Naturalis Biodiversity Center. The funders had no role in study design, data collection and analysis, decision to publish, or preparation of the manuscript.

==============================
Background

Soon after the Spanish conquest of the Americas, the first tomatoes were presented as curiosities to the European elite and drew the attention of sixteenth-century Italian naturalists. Despite of their scientific interest in this New World crop, most Renaissance botanists did not specify where these ‘golden apples’ or ‘pomi d’oro’ came from. The debate on the first European tomatoes and their origin is often hindered by erroneous dating, botanical misidentifications and inaccessible historical sources. The discovery of a tomato specimen in the sixteenth-century ‘En Tibi herbarium’ kept at Leiden, the Netherlands, triggered research on its geographical provenance and morphological comparison to other tomato specimens and illustrations from the same time period.

Methods

Recent digitization efforts greatly facilitate research on historic botanical sources. Here we provide an overview of the ten remaining sixteenth-century tomato specimens, early descriptions and 13 illustrations. Several were never published before, revealing what these tomatoes looked like, who saw them, and where they came from. We compare our historical findings with recent molecular research on the chloroplast and nuclear DNA of the ‘En Tibi’ specimen.

Results

Our survey shows that the earliest tomatoes in Europe came in a much wider variety of colors, shapes and sizes than previously thought, with both simple and fasciated flowers, round and segmented fruits. Pietro Andrea Matthioli gave the first description of a tomato in 1544, and the oldest specimens were collected by Ulisse Aldrovandi and Francesco Petrollini in c. 1551, possibly from plants grown in the Pisa botanical garden by their teacher Luca Ghini. The oldest tomato illustrations were made in Germany and Switzerland in the early 1550s, but the Flemish Rembert Dodoens published the first image in 1553. The names of early tomatoes in contemporary manuscripts suggest both a Mexican and a Peruvian origin. The ‘En Tibi’ specimen was collected by Petrollini around 1558 and thus is not the oldest extant tomato. Recent molecular research on the ancient nuclear and chloroplast DNA of the En Tibi specimen clearly shows that it was a fully domesticated tomato, and genetically close to three Mexican landraces and two Peruvian specimens that probably also had a Mesoamerican origin. Molecular research on the other sixteenth-century tomato specimens may reveal other patterns of genetic similarity, past selection processes, and geographic origin. Clues on the ‘historic’ taste and pest resistance of the sixteenth-century tomatoes will be difficult to predict from their degraded DNA, but should be rather sought in those landraces in Central and South America that are genetically close to them. The indigenous farmers growing these traditional varieties should be supported to conserve these heirloom varieties in-situ.

Introduction

Soon after Christopher Columbus’ first voyage to the Americas, the first New World crops were taken to Europe as curiosities and presented to the royal courts (Pardo Tomás & López Terrada, 1993; Katz, 2009). Seeds of maize, marigold and chili peppers were planted in noblemen’s gardens as exquisite novelties, where they attracted the interest of early sixteenth-century scholars (Daunay, Laterrot & Janick, 2007; Egmond, 2016). One of the American crops that travelled from indigenous farms through the hands of Spanish colonizers to European aristocrats’ gardens was the tomato (Solanum lycopersicum L.). After the conquest of the Aztec city of Tenochtitlan (now Mexico City) in 1521 by Hernán Cortez, the Franciscan friar and chronicler De Sahagún (c. 1577: 49) reported that the Aztecs cultivated a great variety of tomatoes of different sizes, shapes and colors. The Spanish later adopted their Nahuatl term tomatl as tomate (Long, 1995).

The port of Seville was the principal point of entry for products from the New World (Jenkins, 1948; Rotelli, 2018). Still, there is no record of the introduction of the tomato in this Spanish port, or its cultivation in the royal Iberian gardens (Jenkins, 1948), as plant transfers were rarely considered important enough to document (Long, 1995). Due to the many Italian merchants sailing under Portuguese and Spanish flags, and the fact that the Kingdom of Naples was under Spanish rule, these new exotic plants quickly reached Italy (Rotelli, 2018). Soon after the first tomato seeds sprouted in the gardens of Italian aristocrats in the 1540s, they became the object of study by Renaissance naturalists, who described and depicted these ‘golden apples’ with great interest (Daunay, Laterrot & Janick, 2007; Egmond, 2018). From an unknown aphrodisiac to an essential ingredient in national dishes, the subsequent European history of the tomato has been extensively studied (e.g., Sturtevant, 1919; McCue, 1952; Gentilcore, 2010; Metro-Roland, 2013).

Despite their scientific interest in this recently introduced crop, most sixteenth-century botanists did not specify where their tomatoes came from. An exception was the Venetian naturalist Pietro Antonio Michiel, who mentioned that the fruits were known as ‘love apples’ by some and as ‘Peruvian apples’ by others (Poma amoris da alcuni et del Peru, De Toni, 1940). Although Jenkins (1948) classified the latter name as dubious, it gave rise to the alternative hypothesis that the first European tomatoes were brought from Peru, shortly after Francisco Pizarro’s conquest of the Inca emperors in 1531 (Bailey, 1886; Peralta, Spooner & Knapp, 2008).

The geographic origin of tomato domestication has been debated for at least two centuries (Klee & Resende Jr, 2020). Evidence for the ‘South American theory’ was provided early on by the discovery of wild relatives of tomato along the coastline between Ecuador and northern Chile (Jenkins, 1948; Peralta, Spooner & Knapp, 2008). Molecular studies have demonstrated a high genetic and morphological diversity of traditional tomato varieties on the eastern slopes of the Andes in Ecuador and Peru (Blanca et al., 2015; Knapp & Peralta, 2016), but the Mexican origin of the cultivated tomato was still considered (Peralta & Spooner, 2007). The current model for the tomato domestication process is that the small-fruited Solanum lycopersicum var. cerasiforme (Dunal) D.M.Spooner, G.J.Anderson & R.K.Jansen originated from the red-fruited wild species S. pimpinellifolium L, which spread slowly northwards from the Peruvian desert to Mesoamerica, adapting itself gradually to wetter environments, unrelated to human activity (Blanca et al., 2021). Later, indigenous people took the wild Mexican cherry-sized tomato to South America, where it was domesticated, and brought it back to Mesoamerica, where they further domesticated it into the very variable big-fruited S. lycopersicum L. var. lycopersicum (Blanca et al., 2021; Blanca, 2021). Details on the exact time and place of domestication of the tomato are still not known with certainty for either Mexico, Ecuador or Peru (Bai & Lindhout, 2007), but there is a diminishing genetic diversity from Ecuador to Mexico (Lin et al., 2014; Blanca et al., 2015).

In 1989, Sergio Toresella, an expert on medieval herbals, examined a well-preserved tomato specimen in a sixteenth-century book herbarium kept at Naturalis Biodiversity Center in Leiden, the Netherlands. He claimed that this plant collection was made in Ferrara (Italy) between 1542 and 1544 and therefore was the oldest existing herbarium (Toresella, 1992). This meant that the anonymous Italian maker of this ‘En Tibi herbarium’ had collected the earliest European tomato specimen (Houchin, 2010; Thijsse, 2012; Egmond, 2016). As such, the collector would have predated compatriots Pietro Andrea Matthioli, who described a ‘new species’ in his section on mandrake in 1544 (McCue, 1952), and naturalist Ulisse Aldrovandi, who collected in 1551 a specimen of a cultivated tomato, preserved at the Bologna Herbarium, that was considered as the earliest extant specimen (Peralta, Spooner & Knapp, 2008).

The Leiden specimen was also thought to be older than a tomato in a herbarium in Rome, dated pre-1553 (De Toni, 1910), which was attributed first to the painter Gherardo Cibo (Penzig, 1905) and later to the physician Francesco Petrollini (Chiovenda, 1909). However, the ‘En Tibi tomato’, with its simple flowers and round fruit (see https://data.biodiversitydata.nl/naturalis/specimen/L.2111092), did not resemble the well-known sixteenth-century woodcut illustration of a tomato plant with double flowers and elongated, segmented fruits, claimed as typical for the early European tomatoes (Sturtevant, 1919; Daunay, Laterrot & Janick, 2007). This woodcut is often inaccurately attributed to Matthioli (e.g., Houchin, 2010), but was published eight years after his death by Camerarius in his commentaries on Matthioli, first in black and white and four years later in color (Camerarius, 1586: 821; Camerarius, 1590: 378). In the Aldrovandi manuscripts, kept at the University of Bologna, there is an undated list of seeds sent by Aldrovandi to Camerarius that mentions ‘Pomum amoris flore rubro non compressum’ (Aldrovandi manuscripts 136 VII, c. 26).

The finding of the ‘oldest extant tomato’ in the Netherlands led to claims in the popular media that the DNA of this ‘primitive tomato’ could disclose potential ancient resistances to pests and diseases lacking in modern crops. It was suggested that the En Tibi tomato could help plant breeders develop new cultivars with the ‘original taste’ of the sixteenth-century tomatoes (Van Santen, 2012; De Boer, 2013). The genomic diversity stored in herbarium specimens creates ample opportunities for genome-scale population and domestication studies (Staats et al., 2013). Comparing the DNA of traditional crop specimens to the increasingly available online genetic information on crop accessions worldwide can also provide detailed information on geographic origins, past selection processes and historic migration routes of plants and people (Van Andel et al., 2016; Larranaga, Van Zonneveld & Hormaza, 2021). Unfortunately, the sampling of historical collections has had limited success due to their highly degraded DNA, although significant progress is being made with new ‘ancient genomics’ methods (Bakker et al., 2020).

At the same time, ongoing digitization efforts greatly facilitate the research on sixteenth-century herbaria, illustrations, publications and manuscripts (Koning et al., 2008; Van Andel, 2017). However, the literature on early tomato descriptions and depictions often lacks detailed links to the original sources. The latter can now be directly inspected online and sometimes reveal other authors, editions, dates and species than previously thought. Our recent revision of the En Tibi herbarium uncovered that it was not made in Ferrara in 1542-3 as had been suggested, but in Bologna around 1558 by the Italian botanist Francesco Petrollini, who also made the so-called ‘Erbario Cibo’ kept in Rome (Stefanaki et al., 2018; Stefanaki et al., 2019).

This paper aims to provide a more accurate overview of early sixteenth-century descriptions, illustrations and particularly herbarium specimens of the tomato. Some of the published sources have been digitally available for some years, but several images and most of the herbarium specimens have never been published so far. We show that the earliest tomatoes in Europe came in a variety of colors, shapes and sizes, and reveal that some ‘early tomatoes’ were, in fact, misidentified and represent other, related species. We compare these findings with recent molecular research on ‘En Tibi’ specimen’s nuclear DNA (Michels, 2020) and choloroplast DNA (Kakakiou, 2021), which shed new light on its probable geographic origin.

Materials & Methods

We performed a literature review, starting with studies on the introduction of the tomato in Europe (e.g., Jenkins, 1948; McCue, 1952; Daunay, Laterrot & Janick, 2007; Gentilcore, 2010) and on early modern naturalists in Italy, France, Central Europe and the Low Countries (e.g., De Toni, 1907; De Toni, 1910; De Toni, 1940; Findlen, 1994; Findlen, 2017; Egmond, 2016; Egmond, 2018; Rotelli, 2018). We also reviewed modern taxonomic and molecular studies on the origin of the tomato (e.g., Peralta, Spooner & Knapp, 2008; Lin et al., 2014; Blanca et al., 2021). We consequently traced the original sixteenth-century manuscripts cited in these works via online repositories (e.g., google books, the Biodiversity Heritage Library, https://www.europeana.eu).

We searched for tomato specimens in sixteenth-century herbaria (for an overview of the c. 32 herbaria that were probably produced in that century, see Thijsse, 2016) by reviewing scientific studies on these historical collections (e.g., Kessler, 1870; Caruel, 1858; Camus & Penzig, 1885; Penzig, 1905; Speta & Grims, 1980; Soldano, 2000). Where available, we checked the published species lists, and otherwise the indices and specimens of these herbaria, for references to ‘pomo’, ‘mala’, ‘lycopersicon’, ‘Lycopersicum’, ‘Solanum’, etc. We approached several libraries and museums in Italy, France, Germany, Poland and Switzerland to request digital images of specimens and illustrations in manuscripts that had not yet been published. We provided links to digital sources of the historical specimens, literature, manuscripts and images that we reviewed for this study. We listed the local and pre-Linnaean scientific names for tomatoes mentioned in the original published sources, manuscripts, and handwritten texts on botanical vouchers, illustrations or herbarium labels. We checked each historical specimen, description and depiction for visible or written evidence of different shapes, sizes and colors of flowers and fruits. We scrutinized all historical material for possible clues to the geographical origins of the tomatoes. Finally, we report on two recent molecular studies on the genetic affinities of the sixteenth-century tomato specimen in the En Tibi herbarium (Michels, 2020; Kakakiou, 2021).

Results

The first mention of a tomato (1544)

In 1544, the Italian physician and botanist Pietro Andrea Matthioli (1501–1578) was the first person to mention the tomato in Europe, in the first edition of his commentary in Italian on the famous classical herbal De Materia Medica by Dioscorides (c. 60 AD), entitled: ‘Di Pedacio Dioscoride Anazarbeo libri cinque della historia, et materia medicinale trodotti in lingua volgare Italiana’. In his chapter on mandrake (Mandragora), he adds: “Another species [of eggplant, Solanum melongena L.] has been brought to Italy in our time, flattened like the mele rose [a type of apple] and segmented, green at first and when ripe of a golden color, which is eaten in the same manner [as the eggplant: fried in oil with salt and pepper, like mushrooms]” (Matthioli, 1554: 326). Matthioli’s first publication is not available online, so we relied on the translation by McCue (1952). Unfortunately, there is no illustration. The second edition (Matthioli, 1548) had the same text and still did not mention any local name for the tomato. Matthioli’s work became a bestseller, selling over 30,000 copies, and he constantly enlarged the book with augmented editions (Palmer, 1985). In 1554, Matthioli translated his commentary in Latin, expanding his text about the tomato, which he described after the eggplant: “Another species has already begun to be imported, flattened, round like apples, ribbed like melons, at first green, in some plants turning gold and in others red. They are colloquially called pomi d’oro, that is, mala aurea. Eaten in the same way [as eggplant with oil, salt and pepper, like mushrooms. That said by Hermolao]” (Matthioli, 1554: 479). With his phrasing, Matthioli suggested that there were multiple introductions of tomato over a longer period of time, with different colors and shapes. The same text appears after the description of the melanzane (eggplant) in many of the later versions of his book, named pomi d’oro in the Italian and mala aurea in the Latin editions. Unfortunately, Matthioli has never produced or commissioned an image of a tomato during his life (Table 1).

Table 1 Sixteenth-century descriptions, specimens and illustrations of the tomato, ordered by author and chronologically.

Author (birth-death year) Title of source	Year of publication(s) with online links	Representation	Names	
Pietro Andrea Matthioli (1501–1578) Commentaries on Dioscorides	1544, 1548, 1549, 1554, 1557–1560, 1562, 1565, 1568	Text	‘another species’ [of eggplant] (1544, 1548); pomi d’oro, mala aurea (1554 onwards)	
Anonymous Pisa garden catalogue	1545–1548? De Toni (1907: 439)	Text	Thumatulum pomum vulgo dictum rubrum et luteum	
Vincenzo Ferrini (Pisa) to Pier Francesco Riccio (Florence) Letter about showing tomatoes to Cosimo I	31 October 1548 López-Terrada (2015)	Text	pomidoro	
Ulisse Aldrovandi (1522–1605) HerbariumBologna public garden plant catalogue	1551, herbarium vol. 1, fol. 368 1568–1582	Herbarium specimen	Pomum amoris. Mali insani species. Tembal quibusd. Tumatl. seu Pomum amoris quibusdam	
Rembert Dodoens (c. 1517–1585) De stirpium historia… Cruydeboeck	1553: 428, 1554: Part III, chapter 82: 471	UnColored woodcut, text	pomum amoris, pomum aureum, Goldt apffel, guldt appel, pome d’amours poma amoris, gulden appelen	
Leonhard Fuchs (1501–1566). Manuscript, Vienna Codex 11, 122, p. 159 (text), 161 (drawing)	1549–1556 (-1561) Partly published (Meyer, Trueblood & Heller, 1999; Baumann, Baumann & Baumann-Schleihauf, 2001).	Text, watercolor drawing	malus aurea, pomum luteum/rubrum/ croceum, goldt Apffelkraut, pomme d’amour	
Georg Oellinger (1487–1557) Magnarum Medicinae partium herbariae …. (manuscript).	c. 1553, f. 541, 543, 545 Partly published by Luztze & Retzlaff (1949)	Watercolor drawings	Mala aurea seu Poma amoris; Poma amoris maiora Lutea ;	
Conrad Gesner (1516–1565) Historia plantarum (manuscript)	1553 (22 September) p. 37 verso, p. 42 recto	Watercolor drawings	Pomo amoris vel aurea, Goldöpfel, pomi d’oro	
Pietro Antonio Michiel (1510–1576), I Cinque Libri di Piante, vol. 3 (Libro Rosso 1: nr. 46 (illustration possibly by Domenico Dalle Greche)	1550–1576 Partly published by De Toni (1940)	Text, watercolor drawing	Licopersicon Galeni, pomodoro da volgari, melongiana da latini, Poma amoris; Poma del Peru. ‘If I should eat of this fruit, cut in slices in a pan with butter and oil, it would be injurious and harmful to me’ (McCue, 1952)	
Francesco Petrollini Erbario B: Vol. 3, nr.722	before 1553	Herbarium specimen, text (in index)	Malus insana. Mandragorae species. Poma amoris	
Francesco Petrollini En Tibi herbarium	c. 1558: Nr. 294	Herbarium specimen	Puma Amoris	
Anguillara (Luigi Squilerno, 1512–1570) Semplici….	1561: 217 written between 1549–1560	Text	Lycopersico di Galeni Pomi d’oro, Pomi del Perù	
Leonhard Rauwolf (1535–1596), herbarium	1563	Herbarium specimen	Poma aurea	
Ducale Estense (anonymous, herbarium)	1570-1598: nr. 142	Herbarium specimen	Pomi di Ettiopia ouer pomi d’oro	
Mathias De Lobel (Lobelius, 1538–1616) Stirpium Adversaria Nova Plantarum seu stirpium historia Kruydtboeck	1571: p. 108 1576: p. 108 1581: 331-332	Text Text Text, uncolored woodcut	Poma amoris, Pomum aureum, Lycopersiumc quorumdam, an Glaucium Dioscoridis?, Golt opffel, Gulden appelen, Pommes dorées, Gold apel. Memita of the Arabs?, Pommes d’orées, Gold appel	
Melchior Wieland (Guilandinus, 1520–1589), Papyrus…. (1572)	1572: 90-91	Text	Americanorum tumatle Tumatle ex Themistithan	
Hieronymous Harder (1523–1607).	1576-1594	Herbarium specimen, drawing	Solanum marinum alii Poma amoris, Portugalischer nachtschatt	
Andrea Cesalpino (1519–1603) De plantis Libri XVI	1583, lib. IV: 211	Text	Mala insana rotundiora, specie Mali Appii, specie Malii rosei	
Libri Picturati (1565–1569?)	A28.080, A28.080v	Two drawings	Pomme d’amour, pomum amoris,	
Joachim Camerarius the Younger (1534–1598) edited version of Matthioli’s Commentaries	1586: 821 1590: 378-379	Uncolored woodcut Colored woodcut	Poma amoris Goldöpffel, Poma aurea, Amoris poma, Lycopersico, pomme d’amours, pomi d’oro	
Caspar Ratzenberger (1533–1603) Herbarium Vol. 3: 490-PICT0240	1556–1592	Herbarium specimen	Pomidoria, poma aurea, Lieboepffel, Goldoepffel	
Caspar Bauhin (1560–1624) Phytopinax	1596: p. 302-303	Text	Solanum pomiferum fructu rotundo, striato, molli. Poma amoris & Pomum aureum Dodon.	
Caspar Bauhin, edited version of Matthioli’s Commentaries	1598: 761	Text, uncolored woodcut	Citing many names used by others and Poma Peruuiana Anguil[lara]	
Caspar Bauhin (1560–1624)	1577-1624	Herbarium specimen + label B15-075.2A	Solanum pomiferum fructu molli C.B. Aurea mala, Dodo. Poma amoris Lob. Cam. Apud Matth. Tab. Basileae ex horto.	
Caspar Bauhin (1560–1624)	1577-1624	Herbarium specimen + label B15-075.2B_1_	Solanum pomiferum fructu rotundo striato molli, C. Bauh. Lycopersicon Galeni, Anguillar. Poma amoris, Dod. Gal. Lob. Tab. 403. 2. Ex hortulo nostro.	
Casper Bauhin (1560–1624)	1577-1624	Herbarium specimen + label B15-075.2B_2	Solanum pomiferum fructu rotundo striato molli, C. Bauh. Lycopersicon Galeni, Anguillar. Poma amoris, Dod. Gal. Lob. Tab. 403. 2. Ex hortulo nostro.	

But where did Matthioli see his first tomato? According to Ubriszy Savoia (1993), his (former) teacher Luca Ghini (c. 1490–1556) had obtained the seeds from the Venetian patrician and naturalist Pietro Antonio Michiel (1510–1576). Next to his house in Venice, Michiel cultivated numerous exotic plants from faraway places, including the Americas, the Near East and northern Africa. His objective was to spread these botanical novelties among his network, so he sent seeds and sprouts of plants to his friends (De Toni, 1940). Michiel was given the charge to curate the Padua garden from 1551 to 1555 when Luigi Squalermo (1512–1570), better known as Anguillara, was prefect. Anguillara had followed Ghini’s classes and worked in his teacher’s private garden in Bologna, and in 1546 became the first prefect of the Padua garden (Minelli, 2010). In 1543, Anguillara assisted Ghini in amassing materials for the Pisa garden (Findlen, 2017), so it is more likely that Anguillara (and not Michiel) provided Ghini with tomato seeds, also because the Padua garden was founded in 1545 (Palmer, 1985), a year after Matthioli described the first European tomato. Michiel apparently started to expand his Venice garden upon his return from Padua in 1555 (De Toni, 1940).

Ghini taught medical botany in Pisa from 1544 to 1555, where he founded the first university botanical garden supported by the Grand Duke of Tuscany, Cosimo I de’ Medici (De Toni, 1907). Cosimo attempted to import and acclimatize various American plants (Gentilcore, 2010), and Ghini enriched the garden with exotic species and taught his many pupils to press and dry botanical specimens between paper (Findlen, 2017). According to McCue (1952): 292), the Pisa garden catalogue manuscript from 1548 ‘does not include any plant identifiable as the tomato’. However, the inventory of this catalogue brought to light by De Toni (1907: 439) lists a plant named ‘Thumatulum pomum vulgo dictum rubrum et luteum’ (Table 1) and suggests that the catalogue with 620 species could have been started already in 1545.

Matthioli did not travel much after he reached his forties (from 1541 onwards he stayed in the small town of Gorizia, near the current border with Slovenia) and simply sent lists of Dioscoridean plants that he had not yet seen or identified to his colleagues (Palmer, 1985). He often included the knowledge of his fellow scientists or local people in the many editions of his books without citing them (Arber, 1986). Ghini had sent many dried specimens to Matthioli, accompanied by written opinions on their identification (De Toni, 1907; Palmer, 1985). If Ghini had already planted his first tomato seeds in the Pisa garden in 1544 (Ubriszy Savoia, 1993), it was likely his description of the tomato that ended up in Matthioli’s first edition of his Commentaries on Dioscorides in 1544.

The first tomato specimen (1551)

One of Ghini’s best-known disciples was Ulisse Aldrovandi (1522–1605), who became famous for his 16-volume herbarium with over 4,000 specimens kept at the botanical garden in Bologna. The tomato specimen is preserved in the first volume (Fig. 1A), which Aldrovandi started in 1551, and is therefore considered the oldest extant botanical voucher of this New World crop (Table 1). Thorough work has been carried out to trace the origin of Aldrovandi’s specimens, but unfortunately for the tomato specimen this information has not survived (Soldano, 2000). Aldrovandi kept an extensive correspondence with other naturalists. From his letters, we know that around 1551, plants were sent to him by Michiel, then employed in the Padua botanical garden (Minelli, 2010), by Ghini from the Pisa garden (Ubriszy Savoia, 1993) and by his companion and guide in the field Francesco Petrollini (Soldano, 2000; Stefanaki et al., 2019).

Figure 1 All extant specimens of tomatoes in sixteenth-century herbaria, in chronological order.

(A) Ulisse Aldrovandi (c. 1551), Vol 1, p. 368. The pair of leaves at the bottom of the page belong to a Citrullus. Photo credit: University of Bologna. (B) Francesco Petrollini (pre-1553) Photo credit: Biblioteca Angelica, Rome, c.49r, Erbario Cibo B, vol. 3. (C) Francesco Petrollini (c. 1558), L.2111092, ‘En Tibi tomato’. Photo credit: Naturalis, Leiden. (D) Leonhard Rauwolf (1563), Photo credit: Naturalis, Leiden. (E) Caspar Bauhin (1577-1624) B15-075.2A. Photo credit: Herbaria Basel, University of Basel. (F) Bauhin B15-075.2B_1. Photo credit: Herbaria Basel, University of Basel. (G) Bauhin B15-075.2B_2. Photo credit: Herbaria Basel, University of Basel. (H) Hieronymous Harder (1576–1594), Photo credit: Bayerische Staatsbibliothek München, Cod.icon. 3, fol. 140v. (I) Ducale Estense herbarium (1570–1580), Photo credit: Archivio di Stato di Modena. (J) Caspar Ratzenberger (1592), Photo credit: Naturkundemuseum Kassel. Written permission to publish these images is provided in Fig. S3 .

Petrollini, whose birth and death dates remain unknown, also attended classes by Ghini and graduated in Bologna in 1551. Two of his tomato specimens have survived: one in his extensive work herbarium, which is known to have consisted of several book volumes by 1553 (De Toni, 1910) and is kept in the Bibliotheca Angelica in Rome, and one in the En Tibi herbarium (c. 1558) that was made on commission, possibly for the Habsburg emperor Ferdinand I (Stefanaki et al., 2019). The tomato specimen in the Rome herbarium has immature fruits. A separate fruit glued on top of the page, partly destroyed by insects, is an immature eggplant and belongs to another specimen (Fig. 1B). We know that Petrollini graduated two years before Aldrovandi and guided him in his early steps in the field. It is, therefore, likely that he started his work herbarium earlier than Aldrovandi (De Toni, 1910), but the tomato appears only in the third volume. The tomato in the En Tibi herbarium is thus not the oldest preserved tomato specimen in the world, although it is the earliest surviving specimen with (the remains of) a mature fruit (Fig. 1C).

We traced 17 surviving sixteenth-century herbaria in Italy, Germany, France, Switzerland and the Netherlands (Table S1), eight of which contain tomato specimens (Figs. 1A–1J). We have no indication of tomato specimens in other surviving herbaria produced in this time period. The oldest extant herbarium was compiled by Michele Merini, also a pupil of Ghini, in the Pisa botanical garden between 1540–1545. His herbarium is not available online, but its contents were published by Chiovenda (1927), and it does not contain a tomato specimen. Another disciple of Ghini, Andrea Cesalpino, also made a herbarium in the Pisa garden between 1555–1563. Although he mentions the tomato in his De plantis Libri XVI (Cesalpino, 1583), there is no tomato among his vouchers (Caruel, 1858). The first herbaria made in France (by Jehan Girault in 1558) and the Low Countries (by Petrus Cadé in 1566, see Christenhusz, 2004) do not have a tomato specimen either. The second herbarium produced in France, that of the German botanist Leonhard Rauwolf, contains a tomato (Fig. 1D), but this specimen was collected during Rauwolf’s field trip in northern Italy in 1563 (Stefanaki et al., 2021). Tomato specimens are also included in the herbaria Estense (Ferrara, Italy), Bauhin (three specimens; Basel, Switzerland), Ratzenberger (Kassel, Germany) and the Herbarium Vivum of Hieronymus Harder (Ulm, Germany); all these collections have been compiled towards the end of the sixteenth century (Table S1, Figs. 1A–1J).

The first image of a tomato (1553)

Although the tomato was common in Mexico at the time of the Spanish conquest, no images of tomatoes made in the New World exist (Daunay, Laterrot & Janick, 2007). An uncolored woodcut illustration, published in 1553 in a Latin herbal by the Flemish doctor and botanist Rembert Dodoens, can be considered the first image of a tomato (Fig. 2A). A year later (Dodoens, 1554), he published a colored version of the same woodcut (Fig. 2B). Also known under his Latinized name Dodonaeus, Dodoens studied at several universities and travelled to France, Germany and Italy from 1535 to 1546, where he may have seen the tomato for the first time. In 1548, he settled in Mechelen (currently Belgium), then a hotspot of sixteenth-century naturalists, who studied exotic plants in the gardens of local noblemen. In a later edition of his herbal, Dodoens (1583) acknowledged the people who supplied him with plants. One of them, Jean de Brancion, had a beautiful garden with many exotic species, obtained via his extensive international network (Egmond, 2010). In Aldrovandi’s manuscripts, kept at the University of Bologna, there are several lists of seeds sent to De Brancion (Frati, Ghigi & Sorbelli, 1907), of which one, dated 10 January 1571, contains a ‘Pomum pomiferum’ listed just before the eggplant, indicated as ‘Mala insane purpurea’ (Aldrovandi manuscripts 136 V, c. 137v). Another possibility is that Dodoens obtained a tomato plant from the garden of the Antwerp apothecary Pieter van Coudenberghe, created in 1548 and containing more than 600 exotic plants (Vandewiele, 1993).

Figure 2 Published and unpublished 16th century tomato illustrations, in chronological order.

(A) Dodoens (1553), (B) Dodoens (1554), (C) Gesner (1553), image credit: Universitätsbibliothek der FAU Erlangen-Nürnberg, MS 2386, 37v, (D) Gesner manuscript (1553), image credit: Universitätsbibliothek der FAU Erlangen-Nürnberg, MS 2386, 42r, (E) Fuchs (1549–1556–1561), image credit: Österreichische Nationalbibliothek, (F) Domenico Dalle Greche/Michiel (1550–1576), image credit: Biblioteca Marciana, (G) Oellinger manuscript (1553: 541), image credit: Universitätsbibliothek der FAU Erlangen-Nürnberg, MS 2362, 541, (H) Oellinger (1553: 543), image credit: Universitätsbibliothek der FAU Erlangen-Nürnberg, MS 2362, 543, (I) Oellinger (1553: 545), image credit: Universitätsbibliothek der FAU Erlangen-Nürnberg, MS 2362, 545, (J) Libri Picturati (1565–1569) A f(olio) 81r, image credit: Jagiellonian library, (K) Libri Picturati A 28 f(olio) 81v, image credit: Jagiellonian library, (L) De Lobel (1572), (M) Camerarius (1586: 821), (N) Camerarius, 1590: 378), (O) Bauhin (1598). Written permission to publish these images is provided in Fig. S3.

On 22 September 1553, in the same year that Dodoens published the first woodcut, two tomato plants were depicted by the Swiss naturalist Conrad Gesner (Table 1, Figs. 2C–2D). Unfortunately, his Historia Plantarum, a beautiful collection with hundreds of colored plant illustrations, was never published. Gesner had travelled to Italy in 1544, where he met Ghini to study his collections (Findlen, 2017), which provides us with a clue to where he may have obtained his tomato seeds. Later, Gesner (1561) wrote that the tomato was grown by Pieter van Coudenberghe in Antwerp (a possible source of Dodoens’ tomato), by Vuoysselus in Breslau (now Poland) and in German gardens by Joachimus Kreichius in Torgau and in Nuremberg by George Oellinger. Apothecary Oellinger (Ollingerus) also had three drawings made by Samuel Quichelberg (1529–1567) of the different tomato varieties that he had planted (Figs. 2G–2I). His vast collection of naturalist drawings, Magnarum medicinae partium herbariae et zoographiae, was finished in 1553 but never published until Luztze & Retzlaff (1949) published a selection of his work.

In the meantime, from c. 1550 to his death in 1576, the Venetian nobleman Michiel worked on his garden inventory, finalized in a five-volume illustrated manuscript now held by the Marciana library in Venice (De Toni, 1940). Michiel attempted to describe all plants he knew, so the species that figure in his work may have grown in the Padua garden, in his own Venice estate, or they were sent to him as dried specimens (De Toni, 1940). The third volume (Libro Rosso I) features a description of the tomato (Table 1). When he started his manuscript, Michiel was still in Padua and may have seen the tomato there. The watercolor image in this manuscript is possibly made by Domenico Dalle Greche (Fig. 2F). Another drawing in Michiel’s manuscript (Fig. S1) was mentioned as one of the earliest depicted tomatoes in Europe (Egmond, 2018), but the depicted plant has simple, lobed leaves and symmetrical, depressed and deeply furrowed fruits. We agree with De Toni’s identification of this illustration as an Ethiopian eggplant (Solanum aethiopicum L.), probably a member of the kumba cultivar group (PROTA, 2015).

Another candidate for the earliest extant European drawing of the tomato is a watercolor image (Fig. 2E) in a manuscript by the German botanist Leonhart Fuchs, dated between 1549 and 1561 and known as the ‘Vienna Codex’ (Meyer, Trueblood & Heller, 1999; Baumann, Baumann & Baumann-Schleihauf, 2001). This manuscript was meant to become an extended version of his famous herbal De historia stirpium commentarii insignes (Fuchs, 1542), widely considered a masterpiece with 500 very accurate woodcut illustrations and the first known European publication of New World plants such as maize, tobacco, marigold and chili pepper (Meyer, Trueblood & Heller, 1999). The tomato, however, was not yet described in this famous herbal, nor its later editions. It does appear in the Vienna Codex as a drawing (Fig. 2E) and in the text, which reported that this unfamiliar ‘apple’ was only known from gardens and that it was not mentioned by the ancient Greeks, Romans or even the Moors. The manuscript was never published, but Meyer, Trueblood & Heller (1999) suggested that the drawing may be earlier than the woodcut of Dodoens (1553).

In 1586, decades after the first tomato illustrations in the 1550s and eight years after Matthioli’s death in 1578, an uncolored woodcut of a tomato plant (Fig. 2M) appeared in De Plantis Epitome Utilissima, an enlargement of Matthioli’s work published in Latin by Camerarius (1586: 821). A colored version of the same woodcut (Fig. 2N) is published four years later, again by Camerarius, but this time in German (1590: 378), although this image is often attributed to Matthioli (e.g., Houchin, 2010).

The first names of tomatoes

In 1548, Grand Duke Cosimo I was presented some tomatoes from his Florentine Estate. A letter from 1548 mentions that the Florentine pomidoro arrived safely at the ducal household (Table 1). This letter is the earliest written evidence of the term ‘golden apples’ in Italian (Gentilcore, 2010). The Latin translation of this local name (‘mala aurea’) quickly follows in 1554, while Aldrovandi’s name ‘mali insani’ refers to its resemblance to the botanically related eggplant or melanzana (Table 1). Other early sixteenth-century names of the tomato reveal that it came in different colors (red, golden, brown, yellow) or that it was related to the mandrake (‘Mandragorae species’).

The terms ‘pomum amoris’ or ‘pomme d’amour’ are often said to refer to the alleged aphrodisiac properties of the tomato (Smith, 1994). The French term was likely added by a French translator of Matthioli’s work (Peralta, Spooner & Knapp, 2008) and could also be a corruption of ‘pome dei Moro’ (apples of the Moors, Houchin, 2010) or ‘pomi d’oro’. Two years before Matthioli’s first description of the tomato in 1544, the term ‘amoris poma’ was already coined by Fuchs (1542: 532) in his description of the eggplant. Michiel also described the eggplant as ‘Pomes da mouri da Galli, Melongena da Arabi’, a fruit brought by the Moors or Arabs (De Toni, 1940). Solanum melongena L. was indeed introduced to Europe during the Middle Ages by Arab traders from India (Daunay, Laterrot & Janick, 2007).

The Spanish gave the name ‘love apple’ first to the Mexican tomatillo (Physalis ixocarpa Lam.), of which the calyx splits open to reveal the fruit, apparently reminding them of female genitals. Later the Spanish transferred this name to the tomato (Gentilcore, 2010). Although the Italians never adopted the Spanish name ‘tomate’, derived from the Nahuatl ‘tomatl’ (Long, 1995), the appearances of ‘thumatulum’ in the inventory of the Pisa garden, ’tumatl’ in the inventory of the Bologna garden and ‘Tumatle Americanorum’ in Guilandinus (1572), successor of Anguillara in the Padua garden, suggest that some early modern botanists knew this name. However, the local term ‘poma/pomo’ was more common (Table 1).

The name ‘Saliunca’ in the En Tibi herbarium was erroneously given to the tomato specimen, a mistake made by the scribe who wrote the plant names next to the specimens: the name was meant for the preceding specimen (nr. 293) of Valeriana celtica L. (Stefanaki et al., 2018). According to Ubriszy Savoia (1993: 581), Aldrovandi’s term ‘Tembul quibusd.’ (another type of Tembul) refers to Solanum betaceum Cav., the South American tree tomato, but this species was only introduced in European botanical gardens in 1836.

The remark that ‘some people knew the tomatoes as Peruvian apples’ was made both by Michiel (De Toni, 1940) and Anguillara (1561), which is not surprising as they were friends and worked together in the Padua garden from 1551 to 1555 (Minelli, 2010). Several other Andean plants figure in Michiel’s garden inventory (De Toni, 1940), such as coca (Erythroxylum coca Lam.) and ‘quina de India’ (probably Cinchona sp.). As Michiel never published his garden inventory, Anguillara (1561) was quoted for this South American provenance by C. Bauhin in his annotated edition of Matthioli’s commentaries (Bauhin, 1598: 761, Table 1). According to Jenkins (1948), however, there is nothing in the historical record that suggests a Peruvian origin of the tomato. Nevertheless, De Candolle (1885) argued that sixteenth century botanists had received the plant from Peru. De Candolle based this on J. Bauhin & Cherler (1651), published posthumously) who mentioned the name ‘Pomi del Peru’ as a vernacular Italian name. They also mentioned the name ‘Mala Peruviana’, citing Hortus Eystetensis (Besler, 1613) as the source, but this name is not mentioned in the tomato illustration in this book. Gray & Trumbull (1883) assume that Anguillara mistook the tomato for Datura stramonium L., an American Solanaceae described as ‘mala peruviana’ by Guilandinus (1572). Despite his closeness to Ghini, there is no evidence that Anguillara made a herbarium, so no specimen of the ‘Poma del Peru’ exists. In the extensive collection of Aldrovandi’s manuscripts, however, there are many lists of objects (plants, animals, minerals) that he received from all over the world, including South American locations such as the Tumbes province in Peru, the Ecuadorian capital Quito, Cumana (Venezuela) and Uraba in Colombia (Frati, Ghigi & Sorbelli, 1907). It is unfortunately unknown whether Aldrovandi received his tomato specimen directly from his contacts overseas and, if so, from which location. Guilandinus (1572) referred to the tomato as ‘tumatle’, using its Nahuatl name, and wrote that it came from ‘Themistithan’, according to Jenkins (1948) a corruption of Tenochtitlan, the Aztec name for what is now Mexico City. Aldrovandi also made a ‘Themistitani catalogus’ of natural objects received from this area, next to lists of specimens from other Mexican locations such as ‘Iztapalapa’, ‘Jucatan insula’ and ‘Tlaxcala’ (Frati, Ghigi & Sorbelli, 1907: 181). Still, we do not know whether tomatoes are listed in these manuscripts.

The name ‘Ethiopian apple’ written next to the tomato specimen in the anonymous Ducale Estense herbarium (Fig. 1I, Table 1) refers to an African origin. This demonstrates the existing confusion between Solanum lycopersicum and the related Old-World species S. aethiopicum, also depicted in Michiels manuscript (Fig. S1). Besides the tomato specimens, there are also three specimens of S. aethiopicum in C. Bauhins’ herbarium, one of which was named ‘poma amoris racemosa’ and possibly came from his own garden (Fig. S2). The word ‘Ettiopia’ or ‘aethiopicum’ in those days did not refer to the current country of Ethiopia but was used as a general term to indicate the African continent (De Toni, 1940).

The name Lycopersicon means ‘wolf peach’, after the Greek words for wolf (lykos) and peach (persikon), and was first used by the Greek physician Galen (AD 131–200) for designating a plant from Egypt with malodorous sap, just like tomato leaves. Which species Galen had in mind while describing the wolf peach has been lost in centuries of translations and misinterpretations of the classical texts during the Middle Ages (Palmer, 1985). Galen had never seen any New World plant, but a major aim of the Renaissance naturalists was to search for plant specimens that matched descriptions by the classical authors (Palmer, 1985; Stefanaki et al., 2019). However, the German botanist Fuchs argued in his manuscript that as the Greek and Latin authors did not mention the tomato, the plant should not carry any of the classical names (Meyer, Trueblood & Heller, 1999). The Greek name was used in Latin as specific epithet of Solanum lycopersicum L. by Linnaeus (1753), after which Miller (1754) applied it as the genus name for the cultivated tomato (Lycopersicum esculentum Mill). Modern taxonomy has brought the tomato back to the genus Solanum (Peralta, Spooner & Knapp, 2008). Another attempt of sixteenth-century naturalists to trace the tomato in ancient literature led them to the ‘Glaucium’ of Dioscorides: De Lobel (1571) and De Lobel & Pena, 1576) described, not without doubts, the tomato under poppies.

The morphology of early tomatoes

The woodcut illustration of the elongated, segmented tomatoes by Camerarius (1586) and Camerarius (1590) became widely known, as Matthioli’s Commentaries on Dioscorides continued to be a bestseller after his death. However, the sixteenth-century herbarium specimens and the images of small spherical tomatoes in unpublished manuscripts remained locked up in royal treasure rooms, libraries, and universities. This has led to the idea that the earliest tomatoes introduced to Europe were ‘large and lumpy’, a ‘mutation’ from a smoother, more diminutive Mesoamerican form, and probably ‘the direct ancestor of some modern cultivated tomatoes’ (Smith, 1994:15). According to Sturtevant (1919), there were no indications that the round tomato was known among the early botanists before 1700.

From our review of the sixteenth-century descriptions, images and herbarium specimens, it becomes clear that different landraces of tomatoes were introduced early on in Europe. These represented a great variety in flower and fruit shape, size and color, as was already suggested by Daunay, Laterrot & Janick (2007) and Peralta, Spooner & Knapp (2008). Several tomato illustrations (e.g., Camerarius, 1586) and specimens like those of C. Bauhin (Figs. 1E–1G) show duplications of sepals and petals, exserted styles and deeply furrowed (segmented) fruits, while the specimens in the En Tibi and Rauwolf herbaria (Figs. 1C–1D) and Oellinger’s third drawing (Fig. 2I) have simple flowers (five petals) and small, spherical fruits (Table 2).

Table 2 Morphological characters of early sixteenth-century tomatoes mentioned in descriptions or visible in herbarium specimens and illustrations, arranged chronologically.

Author / artist (year)	Collection	Flowers	Fruit shapes	Fruit colors	
Matthioli (1544)	Description	–	‘Segmented’	‘Blood red, gold’	
Aldrovandi (1551)	Specimen	Simple	No fruit	–	
Petrollini (pre-1553)	Specimen	Simple	Small immature fruit	–	
Fuchs (1549–1556/1561)	Description, drawing	Simple and fasciated (‘9 petals’)	Either spherical or oblong, smooth or deeply grooved	Golden, saffron, red, striped, whitish-yellow	
Dodoens (1553)	Description, uncolored woodcut	Fasciated	Ribbed, round, somewhat flattened	Red, yellow or whitish	
Dodoens (1554)	Description, colored woodcut	Fasciated	Ribbed, round, somewhat flattened	Red	
Gesner (1553)	Color drawings	Fasciated, single?	Round and smooth; elongated and ribbed	Red, white, yellow, brown	
Oellinger (1553)	Color drawings	Fasciated and simple	Ribbed and segmented Round and smooth	Red, orange, yellow, whitish?	
Petrollini (1558)	Specimen L.2111092	Simple	Round, smooth	Red	
Michiel/Dalle Greche (1553-1565)	Color drawing	Simple	Spherical, elongated, ribbed, smooth	Red, yellow	
Rauwolf (1563)	Specimen	Simple	No fruit	–	
De Lobel (1581)	Text, uncolored woodcut	Fasciated	Ribbed, round, flattened, ‘big like oranges’	Red, yellow	
Camerarius (1586), Camerarius (1590)	Description, (un)colored woodcut	Fasciated, white	Ribbed/ segmented, elongated	‘Red, golden yellow, brown, some very big’	
Bauhin (1598)	Description, uncolored woodcut	Fasciated, white, yellow	Ribbed, round, flattened, hairy	‘varying in color’	
Harder (1576–1600)	Specimen+drawing	Simple	Round, smooth	Red	
Libri Picturati (1565–1569?)	Drawings	Fasciated	Round, flattened, ribbed	Red	
Cesalpino (1583)	Description	White	Round, elongated and ribbed/furrowed	Golden, red	
Bauhin (1577–1624)	Specimen B15-075.2A	Fasciated?	No fruit	–	
Bauhin (1577–1624)	Specimen B15-075.2B_1	Fasciated	No fruit, label description: ribbed, round, soft	–	
Bauhin (1577–1624)	Specimen B15-075.2B_2	Fasciated	No fruit, label description: ribbed, round, soft	–	
Bauhin (1596)	Description	–	Ribbed, round, soft, some suppressed and wider	Golden yellow (most), some red, pink, white (rare)	
Bauhin (1598)	Uncolored woodcut	Fasciated	Ribbed, round, soft	–	
Ducale Estense (1570–1598)	Specimen	Fasciated	No fruit	–	
Ratzenberger (1556–1592)	Specimen	Fasciated?	Round	Red?	

Although the drawing in Fuchs’ manuscript (Vienna Codex, 1549–1556/1561) is often considered ‘unnatural’ and ‘false’ (Meyer, Trueblood & Heller, 1999; Koning et al., 2008), the task assigned to artist Albrecht Meyer was to represent the variation in flowers and fruits, instead of depicting an individual plant. Fuchs wrote that he had seen at least three different varieties and decided to include all in one illustration (Meyer, Trueblood & Heller, 1999: 629; Peralta, Spooner & Knapp, 2008). Dominico dalle Greche also included several fruit types in his drawing for Michiel (Fig. 2F). According to McCue (1952), the reference by Cesalpino (1583) to the white color of the flowers was incorrect, but Camerarius (1590) described and depicted white-colored flowers as well. The different tomato names, ‘aurea’ (golden), ‘rubrum’ (red), ‘luteum’ (yellow) and ‘croceum’ (orange-yellow, golden-yellow), also indicate that the fruits came in different colors.

Tomatoes underwent a dramatic increase in fruit size during domestication: some modern cultivars produce fruit a thousand times larger than their wild counterparts (Lin et al., 2014). Wild tomato species generally have flowers with five to six sepals, petals and stamens, and bilocular fruits. Through a mutation known as fasciation, flowers will produce up to eight petals and an increased number of locules, which leads to multisegmented, elongated fruits. Humans probably selected fasciated tomatoes for their large fruits, but only a small portion of all modern tomato cultivars is multilocular (Barrero & Tanksley, 2004). The fact that the first tomato described in Europe was segmented (Matthioli, 1544, Table 2) proves that the early sixteenth-century tomatoes did not come from wild plants but represented a crop that had reached a fairly advanced stage of domestication (Bai & Lindhout, 2007).

Table 2 shows that most sixteenth-century specimens lack preserved fruits: juicy tomatoes cannot be easily pressed into botanical vouchers. They are bulky and will not keep their shape when pressed, and due to their moisture, the specimens will quickly start to mold. Petrollini’s first tomato specimen had only an immature fruit, but when preparing the tomato specimen in the En Tibi herbarium, he skillfully removed the juicy insides of the tomato and pressed the skin of the fruit to represent its round shape (Fig. 1C). Ratzenberger’s fruits seem to have spoiled and have been removed from the specimen (Fig. 1J). Harder found a solution: he pressed a flowering specimen and drew the roots, ripe and golden fruits later on the paper (Fig. 1H).

Genetic origin of the En Tibi tomato

What was the geographical origin of the early tomatoes that sparked the interest of the Renaissance botanists? The sixteenth-century literature, specimens and illustrations do not answer this question. The Peruvian origin mentioned by Michiel and Anguillara is not specific, and apart from Guilandinus (1572), the other early sources do not discuss any geographical origin. The knowledge on tomatoes circulating in Europe during the sixteenth-century came from plants that were already cultivated in gardens, as is evident from the detailed morphological descriptions on fruit shape and color, characters that were only observable in living plants. The provenance from the obscure New World was not of interest to most sixteenth-century scholars, who tried hard to trace the tomato in the writings of ancient Greek authors. Regarding herbarium specimens, we only know that the Rauwolf tomato was collected somewhere in N. Italy (Stefanaki et al., 2021), while C. Bauhin’s tomatoes were possibly cultivated in his garden in Basel.

The question of geographical origin may also be approached by genomic research on the crop’s earliest herbarium specimens. Recently, DNA was extracted from a leaf of the tomato specimen in the En Tibi herbarium (c. 1558, Bologna, kept at Naturalis), and its whole genome was sequenced using Illumina TruSeq technology (Michels, 2020) and published online (https://www.ncbi.nlm.nih.gov/bioproject/PRJNA566320; sequencing read archive number SRS5407108). The En Tibi genome was then mapped to the Heinz 1706 reference genome (The Tomato Genome Consortium, 2012), with an average sequencing depth of 2.28 (Michels, 2020). Only 9.9 Mbp were recovered with ≥10x depth, which equated to 1.2% of the reference genome. This indicated that the specimen’s DNA had severely fragmented over the past 475 years. Data on genome assemblies of 114 accessions of wild species and traditional cultivars from Latin America were retrieved from the 360-tomato resequencing project (Lin et al., 2014; https://solgenomics.net/organism/Solanum_lycopersicum/tomato_360) and cropped to span only the 1.2% of the sequenced En Tibi genome with sufficient coverage.

To identify the En Tibi tomato’s nearest neighbors, Michels (2020) performed a network clustering analysis (NeighborNet, Bryant, 2003). Dimensionality reduction analyses were carried out on the remaining SNPs to investigate coarse genetic similarity among the accessions. In Fig. 3, the lengths of the terminal branches are proportional to the number of autapomorphies, distinctive genetic features that are unique to each taxon. Wild populations are generally more genetically diverse (and thus have higher numbers of autapomorphies) than domesticated ones, because of the founder events of domestication and deliberate inbreeding. The highly diverse, wild Solanum pimpinellifolium accessions (dark green circles) spread out on the left (Fig. 3A). On the right, the En Tibi tomato clustered in the group of domesticated tomatoes (S. lycopersicum) from both Central and South America, with very short branches (Fig. 3B). The graph also shows that some accessions of the cherry tomato (S. lycopersicum var. cerasiforme) are genetically close to the large-fruited domesticated tomato varieties on both parts of the continent. In contrast, other accessions of cherry tomatoes appear to be truly wild, given their long branches.

Figure 3 Results of the Neighbor Net clustering analysis, showing the genetic similarity of the wild relatives and the domesticated tomato specimens analyzed by Michels (2020).

(A) Wild individuals of S. pimpinellifolium and S. lycopersicum L. var. cerasiforme from Peru (green circles) and Ecuador (bright green circles) show a high genetic diversity (left of the figure), while a dense cluster of domesticated, genetically less diverse tomatoes is visible on the right, which includes the En Tibi specimen. (B) Enlargement of the cluster with domesticated tomatoes from Fig. 3A, showing the nearest neighbors of the En Tibi tomato (gray circle). All distances expressed in Kimura 2-parameter substitutional distance; parsimony-uninformative SNPs excluded.

Table 3 shows the genetically close varieties to the En Tibi tomato, and some of the associated data stored for these accessions in the C.M. Rick Tomato Genetics Resource Center (TGRC, https://tgrc.ucdavis.edu) at the University of California at Davis, USA. While the three Mexican accessions are characterized as ‘Latin American cultivars’ (probably landraces are meant here), the other three accessions are classified in the TGRC database as ‘wild’. However, C-61 was collected from a family garden and C-281 in open vegetation along a road in the (once) heavily forested eastern Andean foothills. Very little information from the farmers themselves is stored for the accessions close to the En Tibi tomato. B-249 is the only one with a vernacular name (Zocato, no language indicated), and B-153 was collected on a market but said to grow wild. For C-281, the sentence “Indian women: no word in Quechua” in the database suggests that the collector tried to obtain information from a local person, but communication was not possible. The presumably ‘wild state’ of some of the accessions close to the En Tibi tomato does not coincide with the molecular data, which show that the sixteenth-century tomato was a fully domesticated crop. Combined with the absence of farmers’ knowledge in the database, the information in the TGRC database on the domestication status of these accessions is questionable. Some of the nearest neighbors of the En Tibi tomato that were listed as ‘wild’ in the germplasm data may have escaped from cultivation. Compared to genuinely wild accessions, the branches of these presumably feralized ones are so short that they are very likely to have passed through domestication processes and/or possible hybridization with cultivated tomatoes.

Table 3 Tomato landraces close to the En Tibi tomato (c. 1558), in order of genetic similarity.

Identifier Michels (2020)	TGRC nr. (link)	Morphological traits (TGCR database)	Geographical origin (TGCR database)	Collection year	
B-153 big fruits	LA- 1544	Ribbed tomatoes	Mexico: market Xol Laguna, Laguna Encantada, Campeche, Mexico.	1973	
B-249 big fruits	LA-1462	Large fruit, kidney shaped, purple	Merida, Yucatan, Mexico	1971	
C-233 cherry tomato	LA-1218	Small yellow fruit (1-1.5 cm).	Veracruz, Mexico	1969	
C-61 cherry tomato	LA-2670	Large hairy plant, simple flowers, fruits multi-loculed, 2 cm.	Family garden, 19.5 km from San Juan del Oro, Huayvaruni-2, Rio Tambopata, Puno, Peru	1984	
C-281 cherry tomato	LA-1286	Medium-sized, hairy plant, flowers very tiny, fruits various sizes.	0,5 km N of San Martin de Pangoa, Junin, Peru	1970	

Michels (2020) also found that the En Tibi tomato specimen was more heterozygous than all recently collected accessions from Mesoamerica sequenced by Lin et al. (2014), which had a narrower genetic background. This means that the sixteenth-century specimen was less inbred or domesticated than its current counterparts in Mexico. However, some South American domesticated tomatoes had even higher heterozygosity, perhaps due to gene flow between landraces and crop wild relatives (Michels, 2020).

Discussion

Recently, the chloroplast DNA of the En Tibi specimen was completely retrieved at high coverage by Kakakiou (2021). Consequently, the En Tibi plastome was mapped to the chloroplast genome of S. lycopersicum (NC_007898) and haplotype networks were constructed using the Median-Joining (MJ) method and the accessions of the 360-tomato resequencing project (Lin et al., 2014) to reveal the nearest relatives and give clues regarding its origin. The En Tibi specimen was placed in the same node as all Mesoamerican individuals, together with some Ecuadorian and Peruvian accessions of S. lycopersicum var. cerasiforme (Kakakiou, 2021).

The molecular research on the En Tibi tomato does not provide a definite answer to the exact locality of its domestication, and it was impossible to appoint the En Tibi as a direct ancestor of some modern tomato varieties. However, its predecessor likely came from Mesoamerica. The latest study on the origin of domesticated tomatoes (Blanca, 2021; Blanca et al., 2021) proposed a domestication model considering that the wild forms of S. lycopersicum var. cerasiforme from Mexico have travelled with indigenous people to South America, probably as a weed among maize grains, where it hybridized with wild individuals of S. pimpinellifolium. This hypothesis also considered that as part of the domestication process, people then started to select these hybrids and took them back to Mexico, where they generated S. lycopersicum var. lycopersicum with larger fruits. The Peruvian cherry tomato accessions that were close to the En Tibi tomato were probably also cultivated, and carried some Mesoamerican ancestry that could reflect the domestication model outlined by Blanca et al. (2021).

As more than 98% of its genome could not be read, it is impossible to reconstruct complete gene sequences coding for taste or natural resistance to pest and diseases (Michels, 2020), despite anticipation of this earlier (Van Santen, 2012, De Boer, 2013). To reconstruct the ‘original’ flavor, nutritional qualities and adaptations to the (a)biotic environment of sixteenth-century tomatoes, assuming that these tomatoes possessed those traits and that they were lost through intensive breeding for yield in modern cultivars (Klee & Resende Jr, 2020), research should focus on traditional landraces currently grown by small farmers in Central and South America that most resemble historic varieties.

The accessions sequenced by Lin et al. (2014) in the 360-tomato project reflect centuries of human migration and trade, which has caused extensive gene flow between tomato varieties. The information was obtained from online genomic data, and germplasm institutes store very little information on exact localities or morphological, nutritional and agronomical qualities of these accessions or on the farmers that grow them. Moreover, this resequencing project did not capture the entire tomato diversity in the Americas. Increased sampling of landraces in the Andes and Mesoamerica is essential to fully characterize tomato diversity (Knapp & Peralta, 2016). With decreasing crop diversity and the social, economic and ecological challenges faced by small farmers of indigenous descent to preserve their traditional agricultural practices (Knapp & Peralta, 2016; Petropoulos, Barros & Ferreira, 2019), tracing the ‘sisters’ of the En Tibi tomato back to Mexican or Peruvian smallholders’ gardens will be difficult. The landraces that were genetically close to the En Tibi tomato were collected between 36 and 52 years ago: they may have already disappeared from indigenous gardens and survive only as seeds in germplasm institutes.

Conclusions

The earliest tomatoes that reached Europe came in a wide variety of colors, shapes and sizes: with both simple and fasciated flowers, round and segmented fruits. The first description of a tomato was published by Matthioli in 1544, while the oldest specimens were collected by Aldrovandi and Petrollini in c. 1551 in the Pisa botanical garden. The earliest illustrations were made in Germany and Flanders in the early 1550s. The names of early tomatoes in contemporary manuscripts suggest both a Mexican and a Peruvian origin. The ‘En Tibi’ specimen was collected by Petrollini around 1558 and thus is not the oldest extant tomato, although it is the first specimen that shows a mature fruit. Although only 1.2% of its nuclear DNA was recovered, molecular research on its genome and plastome shows that the En Tibi specimen was a fully domesticated tomato, and genetically close to three Mexican landraces and two Peruvian tomato accessions that most probably also had a Mesoamerican origin.

Molecular research on the other sixteenth-century tomato specimens may reveal additional patterns of genetic similarity and geographic origin. Clues on the ‘historic’ taste and pest resistance of the sixteenth-century tomatoes are difficult to find in their degraded DNA, but should rather be sought in those landraces in Central and South America that are genetically close to them. The indigenous farmers growing these traditional varieties should be supported to conserve these heirloom varieties in-situ.

Sequencing the ancient DNA of the other nine sixteenth-century tomato specimens highlighted in our paper may provide different but equally exciting snapshots of historic genetic variation. This may lead to different, similar-looking landraces in either South- or Mesoamerica. Further digitization, translation and online publication of Aldrovandi’s manuscripts, archives of botanical gardens and correspondence between Renaissance naturalists will probably reveal more details on the first New World crops in Europe, their geographic origin and arrival date.

Supplemental Information

Supplemental Information 1 All extant sixteen-century herbarium in Europe, with their locations, dates and link to tomato specimens if present

Click here for additional data file.

Supplemental Information 2 Drawing of Solanum aethiopicum in the manuscript of Pietro Antonio Michiel (1510-1576)

Artist unknown, manuscript: I cinque libri di piante. Codice Marciano, 1551-1575. Image credit: Biblioteca Marciana, Venice.

Click here for additional data file.

Supplemental Information 3 Specimen of Solanum aethiopicum, collected by Caspar Bauhin between 1577 and 1624

Image credit: University of Basel.

Click here for additional data file.

We would like to thank the staff of libraries and universities who provided us with digital images of specimens, illustrations, rare books and manuscripts: Adriano Soldano, Annalisa Managlia, Martina Caroli and Silvia Tebaldi of the University of Bologna (Aldrovandi herbarium), Gerda van Uffelen of the Hortus Botanicus Leiden and Izabela Korczyńska of the Jagiellonian library in Krakow (Libri Picturati), Gisela Glaeser of the Universitätsbibliothek der FAU Erlangen-Nürnberg (Oellinger and Gesner illustrations), Raffaella Alterio and Mario Setter of the Biblioteca Angelica (Erbario B), Rome, Karien Lahaise, Naturalis library (literature), Jurriaan de Vos, University of Basel (C. Bauhin herbarium), the staff of the Archivio di Stato di Modena (Ducale Estense herbarium), the staff of Biblioteca Marciana, Venice and Alessandro Moro (Michiel’s illustrations), Peter Mansfeld, Naturkundemuseum Kassel (Ratzenberger herbarium), Sophie Schrader, Bayerische Staatsbibliothek München (Harder herbarium), and Peter Prokop of the Österreichische Nationalbibliothek, Vienna (Fuchs illustration). DNA extraction was carried out by Barbara Gravendeel (Naturalis) and sequencing was carried out by Elio Schijlen (Wageningen University). We are also grateful to José Blanca who shared his group’s latest findings on the history of tomato domestication.

Additional Information and Declarations

Competing Interests

Author Contributions

Data Availability

The authors declare there are no competing interests.

Tinde van Andel and Anastasia Stefanaki conceived and designed the experiments, analyzed the data, prepared figures and/or tables, authored or reviewed drafts of the paper, and approved the final draft.

Rutger A. Vos conceived and designed the experiments, performed the experiments, analyzed the data, prepared figures and/or tables, authored or reviewed drafts of the paper, and approved the final draft.

Ewout Michels conceived and designed the experiments, performed the experiments, analyzed the data, prepared figures and/or tables, and approved the final draft.

The following information was supplied regarding data availability:

The links to the digitally available specimens, to the historical books and to the germplasm data are available in the Supplemental Files.

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
