# Peer review of "Sixteenth-century tomatoes in Europe: who saw them, what they looked like, and where they came from"

_PeerJ, doi:10.7717/peerj.12790_

## Round 0.1 · original submission · Major Revisions

The paper is appreciated by the reviewers (and myself) as an intriguing approach to clarify the early types of tomato cultivated in Europe. However it needs improvement, in particular I agree that the genomic study Michels (2020) cannot be found, reviewer 1 is suggesting how to improve this point which is relevant.

Reviewer 1 ·

Basic reporting

See Additional Comments

Experimental design

See Additional Comments

Validity of the findings

See Additional Comments

Additional comments

Review of Sixteenth-century tomatoes in Europe: who saw them, what did they look like, and where did they come from?

Van Andel et al. present a very valuable study to the origin of tomatoes in Europe, using an approach that leverages the few surviving 16th-century herbaria. Specifically, the paper aims (l 142 143) to provide a more accurate overview of early sixteenth-century descriptions, illustrations and herbarium specimens of the tomato. This is a much needed effort, because discussions on tomato origins span well over a century, and this literature is rather littered with incomplete, incorrect or contradictory information. Also, this must have been rather challenging given the diverse array of languages of the relevant sources. In addition, the study presents a fraction of an ancient tomato genome (ca. 1.2%), which the authors downplay as being not very important due to it being so fragmented.

Overall, I am quite enthusiastic about this paper. The topic is highly relevant, the methods are suited to address the questions, the historic research is executed with great care and comes across as rather comprehensive. I also appreciate the effort the authors took in presenting this study rather well-readable; the structure of the Results here certainly helped. The overall result, that the early tomatos in europe were already quite diverse in traits, and point to both peruvian and mexican heritage, is convincing.

Nonetheless, I have a few comments, which I trust the authors will be able to address.

My main criticism is the presentation and the interpretation of the ancient tomato genome.
- Foremost, the authors continuously refer to a study Michels (2020) which is not cited in the literature. So either the ancient genome is already published, in which case it should not be presented as Results of this paper, or it is not yet published, in which case many relevant methodological details are missing (see below).
- Also the rather sweeping interpretation of the ancient genome is a bit surprising. In particular, although the authors state further ancient genomes may generate exciting results (l 601), they also state rather strongly that clues on historic traits of tomatos should not be sought in ancient genomes but in extant "ancient-like" cultivars and landraces. As written, it comes across as dismissing ancient-genomic research altogether (possibly because their result of 1.2% genome is rather modest?) which is probably not what the authors intended.
- Finally, the topic of ancient genomics should be properly separated in Methods, Results and Discussion, which is now all presented in Results.

Some examples of missing or confusing information on the ancient genomics part:
- where is the genome made publicly available?
- Illumina TruSeq is a library preparation method, not a sequencing platform
- what DNA sequencing method and library prep method was used?
- what mapping approach was used?
- how was it established that the sequencing results were authentic, ancient DNA and not any form of contamination (e.g. signs of cytosine deamination at the end of sequences?)
- how was heterozygosity computed to conclude that the sample was "quite heterozygous"?
- genetic diversity of a population does not necessarily lead to many autapomorphies in an exemplar sequence (autapomorphies say something about allele divergence which is a function of time, mutation rate and population size, not about genetic diversity which simply counts the number of different alleles across individuals of a population. They may well coincide but do not need to necessarily).
--> because all these questions remain, if the ancient genome has not yet been published, this part of the manuscript should be re-assessed.

Overall, I would recommend to deal with the ancient genomics more in a hypothesis testing presentation, for instance something along the lines of: "if established ideas that tomatos were introduced from mexico, our ancient genomes closest sister groups should be more closely related to landraces from mexico than from south america" in the introduction. Right now, it comes across as "bolted on" to an otherwise historic herbarium and literature study.

Another general remark concerns Aldrovandi:
l 243-244 states that no information of the origin of tomato specimens of Aldrovandi survive, while l 396-402 speaks of "extensive collection of Aldrovandi's manuscript" that might contain more information but that could not be verified. These two bits are in conflict: so what do we really know about where Aldrovandi had his specimens from? Why could these sources not be verified? Beyond the scope of this study? That would of course be understandable, but it would help to be explicit in the text.

Some more, all minor line by line comments:

l 82 - 95. Somewhat incomplete; especially works of Razifard et al. seem critical here.

l 228. It would be helpful to state where Matthioli spend his days later in life.

l 263. Please indicate whether you suspect the 17 to be the "final number".

l 304. "admire" may be sub-optimal as I suspect Gesner was primarily interested in studying the collections?

l 352 "has no name". This formulations make it seem odd; it would help to be more explicit regarding Matthioli's words, as in l 185-186 (or refer to that bit explicitly.

At various places in the introduction (l 76, 77, 103), early names for the tomato are translated. It would help to also state them literally in the original language.

Figure 1 and 2 are not particularly aesthetically arranged.
The color coding in Fig 3 is a bit unclear because it is rather small. Also, it should be made more clear where the new specimen is located (arrow?).

·

Basic reporting

I consider this contribution to be within the standards required by PeerJ.
In this review I am contributing with my comments and suggestions, which I have also been indicated in the revised document (see attachment)
The article has been written in a technically correct way, although I think some paragraphs are very long and could be summarized.
The texts and bibliographic citations in Latin should be indicated in italics, as well as scientific names. It is necessary to review the entire manuscript, including the Bibliographic list, Figures and Tables.

The authors have conducted an extensive search of the literature published by the first European herbalists in the sixteenth century, as well as of researchers who have investigated the collections of specimens and historical botanical documentation, which contribute to illustrate the initial knowledge of tomatoes grown in Europe.
Some sentences should include the bibliographic citations that support these concepts. See comments in the text:
Line 19 and 55. Could you provide the source to support this statement that first tomatoes were presented to Iberian Kings in Europe?
Line 370. Could you provide a cite for the statement that the Spanish gave the name ‘love apple ’ first to the Mexican tomatillo?
Check the citations in the text, there are missing references in the bibliographic list. For example, Michels, 2020

The article has been structured in an acceptable format and that allows the monitoring of the results in each of the sections, mainly according to a chronological order.
I suggest that Figure 1, which includes sixteenth-century tomato specimens found in different European herbaria, can be improved to achieve greater sharpness and illumination of the images (e.g. from Fig. 1, I). I also suggest cropping the images (e.g. Fig. 1 J where a part comprises the dark sheets of the book) and also standardizing the image formats (dpi) and sizes of each specimen, trying to achieve a similar aesthetic. I also suggest that the letters of each specimen be placed in the lower right corner.
I recommend the same for Figure 2, which includes the illustrations of the cultivated tomato published in the sixteenth century. In this case it is also desirable to improve the sharpness and lighting of the images, crop the dark parts, standardize the sizes of each illustration and organize an harmonic location of the sheets. In the current Figure the letters have been placed in different places, and I suggest that the letters of each specimen could be located in the lower right corner.
In Figure 3 it is necessary to explain in detail the meaning of colors and codes in each circle. In order to understand this scheme, I suggest to complete Table 3 indicating for each species (S. pimpinellifolium and S. lycopersicum L. var. cerasiforme),the accessions and corresponding codes, the place of collections, and other relevant data. This Table may be included as supplement, or indicate where this informtion could be found (previous publications?)

Experimental design

This article presents an interesting approach to elucidate the origin of the first tomatoes grown in Europe. An exhaustive search of the descriptions and illustrations of the sixteen century Herbalists, and the oldest European herbarium specimens was carried out. Through this evidence it is concluded that the first cultivated tomatoes in Europe presented diversity of fruit size, shape, and color. Confusing interpretations of descriptions and illustrations that do not depict tomatoes were also clarified.
Another relevant aspect of this work is the comparison of molecular information obtained from an ancient specimen with a large dataset on genome assemblies of 114 accessions of wild species and traditional cultivars from Latin America. Although a very low percentage (1.2% of the sequenced genome) was obtained from En Tibi anciente tomato specimen, its coverage was sufficiente for comparison and genetic diversity analysis.
Methods and analitical approaches were adecuate and described with suficient details to be replicate.

Validity of the findings

Although, the authors could not elucidate with this approximation the origin of the old specimen cultivated in Europe in the mid sixteenth century and its relationship with a specific geographic area of Latin America, I considered that the greatest contribution of this work is to demonstrate the usefulness of herbarium specimens not only to recover the botanical history but also for modern genomic and diversity studies in tomatoes.
These results are promising for further research of ancient herbarium specimens, exploring old hiden collections, and also encourage to improved DNA extraction techniques.

---

## Round 0.2 · Minor Revisions

To finalize this very interesting paper the authors are left with minor revisions suggested by the reviewers, including rev.2 who is introducing them in the attached file. Best wishes

Reviewer 1 ·

Basic reporting

Review of "Sixteenth-century tomatoes in Europe: who saw them, how they looked like, and where they came from", submitted by van Andel et al. to PeerJ

This is the second time I see this manuscript, and I am positively struck by the care and attention to detail the authors had in revising the manuscript based on the many comments they received. In my opinion, especially the integration of the genetic information is much more satisfactory. Overall, I am rather positive about this paper, and the many other adjustments also improved the paper greatly.

Only a few minor comments:

Abstract: Although I am sympathetic toward the idea of rebutting overly speculative claims in the media (regarding reviving this extinct tomato), I don't think it is necessary to do this in the abstract of this paper.

l 255-258. It strikes me that this phrasing suggest there were recurrent introductions of tomato over a prolonged period of time, due to the use of the tense "has already begun to be imported".

l 289, if there is an estimate of the total number of 16th century herbaria that survive, perhaps in Thijsse 2016, it would be helpful to include it here, whether you could study it or not.

l 543-544. It recommend to state the bioproject number PRJNA566320 and sequencing read archive number SRS5407108 to find the associated reads.

lines 609-615, this seems to be better as a paragraph in the same section as the results that Michels 2020 produced, rather than discussion.

Experimental design

no comments

Validity of the findings

no comments

Additional comments

no comments

·

Basic reporting

I consider this contribution to be within the standards required by PeerJ.
In this second review I am contributing with my comments and suggestions, which I have also been indicated in the revised document. Please consider my suggestions in the final version.
The article has been written in a technically correct way, although I consider some paragraphs are very long and could be summarized (see my comments in the manuscript).
In the literature references include the cittation of Miller (1754) and Mathiolli (1583)

Experimental design

Described in the first review.

Validity of the findings

Explained in the first review.

Additional comments

The author responded the questions and observations made in the first review, and incorporate the required changes to improve the final text.
Please consider my suggestions indicated in this second review of the original manuscript.

---

## Round 0.3 · accepted · Accept

The authors have replied to the reviewers suggestions and modified the manuscript accordingly.

The Section Editor added: I strongly suggest that the title gets fixed - 'what they looked like', not 'how they looked like'.